# Protective Effects of Hydrolyzed Chicken Extract (Probeptigen®/Cmi-168) on Memory Retention and Brain Oxidative Stress in Senescence-Accelerated Mice

**DOI:** 10.3390/nu11081870

**Published:** 2019-08-12

**Authors:** Ming-Yu Chou, Ying-Ju Chen, Liang-Hung Lin, Yoshihiro Nakao, Ai Lin Lim, Ming-Fu Wang, Shan May Yong

**Affiliations:** 1Department of Food and Nutrition, Providence University, Taichung 43301, Taiwan; 2Quanzhou Preschool Education College, Quanzhou 362000, China; 3Scientific Research and Applications, BRAND’S Suntory Asia, Singapore 048423, Singapore

**Keywords:** SAMP8, memory, chicken meat hydrolysate, antioxidant, hippocampal gene expression, anti-aging

## Abstract

The senescence-accelerated prone (SAMP8) mouse model shows age-dependent deterioration in learning and memory and increased oxidative stress in the brain. We previously showed that healthy subjects on a six-week supplementation of a chicken meat hydrolysate (ProBeptigen®/CMI-168) demonstrated enhanced and sustained cognitive performance up until two weeks after the termination of supplementation. In this study, we investigate the effect of ProBeptigen on the progression of age-related cognitive decline. Three-month old SAMP8 mice were orally administered different doses of ProBeptigen (150,300 or 600 mg/kg/day) or saline daily for 13 weeks. Following ProBeptigen supplementation, mice showed lower scores of senescence and improved learning and memory in avoidance tasks. ProBeptigen treatment also increased antioxidant enzyme activity and dopamine level while reducing protein and lipid peroxidation and mitochondrial DNA damage in the brain. Microarray analysis of hippocampus revealed several processes that may be involved in the improvement of cognitive ability by ProBeptigen, including heme binding, insulin growth factor (IGF) regulation, carboxylic metabolic process, oxidation–reduction process and endopeptidase inhibition. Genes found to be significantly altered in both ProBeptigen treated male and female mice include *Mup1*, *Mup17*, *Mup21*, *Ahsg* and *Alb.* Taken together, these results suggest a potential anti-aging effect of ProBeptigen in alleviating cognitive deficits and promoting the antioxidant defense system.

## 1. Introduction

Aging is an inevitable process that may be accompanied by neurodegeneration [1,2,3] leading to cognitive decline [4,5,6]. Under normal physiological conditions, balance exists between free radicals and antioxidants. This balance breaks down with the process of aging, as the generation of radicals overpowers the antioxidant defense system and gradually diminishes its efficacy [7,8]. There are a few hallmarks of brain aging, including a decreased antioxidant defense, increased oxidative stress and an impaired mitochondrial oxidative phosphorylation (OXPHOS) system leading to mitochondrial dysfunction [9]. Oxidative stress has been strongly implicated in cognitive decline during aging [10] as age-related memory loss is often accompanied by the accumulation of oxidative damage to lipids, protein and nucleic acids, resulting in disrupted neuronal function [11]. Mitochondrial dysfunction plays a major role in both the physiological aging process and neurodegeneration [12,13] by generating reactive oxygen species (ROS). This concept has been termed the free radical theory of aging. ROS can lead to oxidative stress induction and damage accumulation such as through DNA mutation or lipid and protein dysfunction [9,14], all of which may contribute to age-related neurodegenerative disorders [15].

The senescence-accelerated mouse (SAM) was first established as an experimental model of accelerated aging by Takeda et al. [16]. SAM comprises of nine substrains of senescence-accelerated-prone (SAMP1, P2, P3, P6, P7, P8, P9, P10 and P11) and three substrains of senescence-accelerated-resistant (SAMR1, R4 and R5) mice showing normal ageing. SAMP substrain SAMP8 has a shorter median lifespan of 9.7 months compared to SAMR1 mice (16.3 months) [17] and is a specific model of memory dysfunction. It shows age-dependent deterioration in learning and memory abilities at an early (2–4 months) age [18,19] with lower incidence of other phenotypic aging compared to other SAMP mice strains [20]. Learning and retention deficits have been well-established in the SAMP8 mice using various types of behavioral paradigm including passive avoidance, one way active avoidance, aversive T-maze, object recognition, lever press appetitive task, Greek cross and Morris water maze [20,21,22,23,24,25]. Moreover, the aged SAMP8 mice exhibit abnormal circadian rhythm and increased oxidative stress compared to age-matched SAMR1 controls [24,26]. The behavioral alteration of the former is thought to be related to increased free radical production in the central nervous system [27,28,29,30] associated with mitochondrial dysfunction [31] observed in the SAMP8 mice as early as two months of age [32]. A wide variety of compounds with anti-oxidative properties have been reported to be able to prolong life span, improve learning and memory disorders and reverse the measures of oxidative stress in SAMP8 mice [14,18,19,33,34,35,36,37]. All these findings suggest that oxidative stress plays a role in the memory dysfunction of SAMP8 mice.

Essence of chicken (EOC) is a concentrated aqueous extract of chicken meat with a long history of consumption in Asia. It is very low in fat, but rich in proteins and peptides, trace elements, carnosine, anserine and amino acids. Previously, studies on the essence of chicken have demonstrated benefits in cognitive performance, particularly in the area of working memory, attention and reaction times [38,39,40]. A recent study showed that EOC exerted anti-fatigue effects in mice during exercise possibly due to its anti-oxidant properties in inducing activities of muscle and liver superoxide dismutase (SOD), hepatic catalase (CAT) and glutathione (GSH) production [41].

A chicken peptide extract called chicken meat hydrolysate (ProBeptigen®/CMI-168) was further enriched from EOC via hydrolysis. A pilot study on a group of healthy middle-aged subjects with a six-week-supplementation of ProBeptigen (670 mg/day) showed enhanced cognitive performance, particularly in learning and memory [42]. These cognitive benefits were sustainable even two weeks after termination of supplementation. More recently, ProBeptigen supplementation in middle-aged mice showed improvement in learning and memory, independent of structural or functional changes to the hippocampus [43]. The mechanism underlying the effect of ProBeptigen on learning and memory still remains to be elucidated.

Antioxidants have been found to prevent and reverse learning and memory deficits induced by free radicals and improve learning in aged rats [44,45,46,47,48,49]. Natural products such as onion and mulberry extracts, lotus seedpod proanthocyanidins (LSPC), oolong and green teas were able to reverse memory impairment, decrease lipid hydroperoxide and elevate antioxidative enzyme activities in brain and serum of SAMP8 mice [14,33,34,35]. Protein hydrolysates containing amino acids such as Glu, Asp, Tyr, Trp, Met, Lys, Cys and His display antioxidant capacity [50,51,52,53,54]. In the free form, these amino acids are able to cross the blood–brain barrier (BBB) [55,56,57,58]. Specifically, chicken breast protein hydrolysate has been shown to have in vivo antioxidant activity in D-galactose-induced aging mice [59]. Since ProBeptigen is a chicken meat hydrolysate, we hypothesize that it may exhibit strong anti-oxidative properties that contribute to its cognition-enhancing effect.

Previous studies have demonstrated that age-related cognitive deficits are accompanied by changes in the gene expression profile in the brain, whether it is normal or senescence-accelerated prone (SAMP) mice [29,60,61]. The hippocampus is one of the brain areas most affected by aging and memory decline. Earlier findings have reported extensive changes in hippocampal genes of SAMP8 mice with aging or compared with SAMR1 normal aged mice, which can be functionally categorized into oxidative stress, neuroprotection, signal transduction, energy metabolism and immune response [29,62,63,64,65]. All of these processes are involved in learning and memory, suggesting that abnormalities of these systems may contribute to the cognitive deficits in SAMP8 mice. Thus, we wanted to investigate the effect of ProBeptigen on the gene expression profile of SAMP8 mice.

In this study, we used the SAMP8 mouse strain as it mimics the deterioration of learning and memory in the elderly and serves as a good model for studying aging and age-induced impairment. We investigated whether supplementation of ProBeptigen could protect the brain from cognitive deficits and promote the brain’s antioxidant capacity in SAMP8 mice. The effect of ProBeptigen on the whole genome microarray-based transcriptome profiling of hippocampi of SAMP8 mice was also examined.

## 2. Materials and Methods 

### 2.1. Subjects and Food Intake

The SAMP8 mice (three-months old) mice were previously established [29] and purchased from the Animal Center of Providence University, Taichung, Taiwan. The study protocol was carried out in strict accordance with the recommendations in the Guide for the Care and Use of Laboratory Animals of the National Institutes of Health and approved by the Animal Research Ethics Committee at Providence University (IACUC number 20170512-A03). All animals were acclimatized for one week with access to food and water ad libitum and maintained at 12-h light/12-h dark cycle at room temperature 25 ± 2 °C and 65% ± 5% humidity.

### 2.2. Animal Treatment and Tissue Collection

In a previous pilot clinical trial, the daily doses of ProBeptigen for an adult human were 670 mg/day [42]. The doses for mouse was converted from a human equivalent dose (HED) based on the following formula from the US Food and Drug Administration (http://www.fda.gov/downloads/Drugs/GuidanceComplianceRegulatoryInformation/Guidances/ucm078932.pdf), assuming a human weight of 60 kg, the HED for 670 mg/60 kg = 11.167 × 12.3 = 137.15 mg/kg (adjusted to 150 mg/kg per day for mouse). The animals were randomly divided into four groups with eight mice per group for each gender. SAMP8 control group was fed with normal chow diet only. The other three groups were orally administered ProBeptigen daily at 150 mg/kg (1×), 300 mg/kg (2×) and 600 mg/kg (4×), respectively. The total duration of the experiment was 13 weeks. The mean daily intake of food and water was monitored and body weight was recorded weekly. After 11 to 13 weeks of supplementation, all animals were scored for degree of senescence and subjected to memory behavioral tests including the open field test, passive avoidance and active shuttle avoidance tasks. After behavioral tests, mice were sacrificed as per protocol. Brain tissues were quickly dissected from the mice, cryopreserved using liquid nitrogen and stored at −80 °C. The tissues were prepared individually accordingly to the manufacturer’s specifications for each assay.

### 2.3. ProBeptigen/CMI-168 Preparation

Chicken meat ingredient (ProBeptigen®/CMI-168) is a hydrolyzed extract prepared from chicken meat using a proprietary processing technology. Chicken meat is subjected to high temperature, high pressure and enzyme hydrolysis to yield a hydrolysate rich in amino acids and peptides. Nutritional content of ProBeptigen is listed in Table 1.

### 2.4. The Degree of Senescence in Samp8 Mice

The total grading score of senescence in six-month-old SAMP8 mice was evaluated according to the grading score system used by Takeda et al. (1981) [16]. Briefly, the parameters consisted of clinical signs and gross lesions associated with behavior, skin, eyes and spine. The eight items include: (1) Reactivity, (2) passivity, (3) glossiness, (4) coarseness, (5) hair loss, (6) skin ulcer, (7) periophthalmic lesion and (8) lordokyphosis. The degree of senescence in each category was graded from 0–4 corresponding positively to the degree of symptoms. For instance, grade 0 represents no particular changes whereas grade 4 represents the most severe changes. The total grading score is the sum of the degrees for the eight items.

### 2.5. Memory Behavioural Tests

#### 2.5.1. Open Field Test (Locomotion)

We examined the activity in an open field to eliminate the possibility that any differences in the acquisition of the avoidance response may be due to changes in locomotion of the treated mice. Mice were individually placed in a cubic box with a length of 25 cm and the locomotor activity of all the mice was measured for 10 min using a video path analyzer (Coulborn Instruments model E16-21).

#### 2.5.2. Passive Avoidance Test

Both the passive avoidance and active shuttle avoidance tests were performed as described previously [14]. Briefly, each animal was placed in a box (35 cm × 17 cm × 20 cm, width × length × height, model E10-15, Coulbourn Instruments, Philadelphia, PA, USA) consisting of two equal compartments connected by a small opening (7.5 cm × 6.5 cm, Guillotine Door, model E10-15GD, Coulbourn Instruments, Philadelphia, PA, USA). One compartment was lit and the other darkened by a black semi-transparent plastic cover. The floor of the box consisted of parallel steel metal rods at 1 cm intervals with electric currents running through the lines.

In the acquisition trial, each mouse was placed in the bright compartment, and after a brief orientation period (10 s), the gate was opened to allow the mouse to enter the dark compartment. When the mouse entered the dark compartment, the gate would close and an electric foot shock (100 V, 0.3 mA, 2 s) would be simultaneously given. Five seconds after the electric shock, the mouse is removed from the dark component and returned to the feeding cage. If the animal does not enter the dark compartment in 300 s, the mouse is forced into the dark component. The gate is closed, an electric shock administered and the mouse is returned to the feeding cage. The retention test was performed on the mice after 24, 48 and 72 h with the retention latency time measuring the time the mice stayed in the bright compartment before entering the shocked-paired dark compartment.

#### 2.5.3. Active (Shuttle) Avoidance Test

The active (shuttle) avoidance test was controlled by a computer program that generated the sound, light and electric shock. Each mouse was allowed a 10-s brief orientation period in one compartment while a conditional stimulus (CS) consisting of 10 s of the stimulus of the tone and yellow light were generated. Following this stimulus, an unconditional stimulus (UCS) consisting of a 0.3-mA, 5-s scrambled foot shock was carried out if the mouse had not escaped by entering the other compartment during the CS. Each mouse was exposed to five CS per session and four sessions per day for a total of 20 trials. The experiment was performed for four consecutive days with animals allowed to rest for 15–20 min between sessions. Successful avoidance responses signify crossing over to the other compartment while trials where the mice do not avoid or escape the foot shock are considered failures. Success on the first day of test meant acquisition of successful avoidance and avoidance responses on the other days signified retention. The avoidance responses of the mice were recorded automatically.

### 2.6. Assessment of Antioxidant Enzymes

Whole brain tissues were promptly dissected after sacrifice and perfused with 50 mM (pH 7.4) ice-cold phosphate buffer saline solution (PBS). Samples were homogenized in cold PBS buffer containing protease inhibitors. Homogenates were centrifuged at 10,000× *g* for 10 min and the supernatants were collected and stored at −80 °C for determining brain superoxide dismutase (SOD), catalase (CAT), glutathione peroxidase (GPx), glutathione reductase (GSH-Rd) and glucose-6-phosphate dehydrogenase (G6PDH) activities.

#### 2.6.1. Superoxide Dismutase Assay (SOD)

Superoxide dismutase (SOD) activity was measured with a Superoxide Dismutase Assay Kit (Catalog No. 706002, Cayman Chemical, Ann Arbor, MI, USA). SOD is a metalloenzyme that catalyzes the dismutation of the O_2_^.-^ to molecular O_2_ and H_2_O_2_ and thus forms a crucial part of the cellular antioxidant defense mechanisms. The activity was measured by recording absorbance at 440–460 nm. One unit of SOD is defined as the amount of enzyme needed to exhibit 50% dismutation of the superoxide radical.

#### 2.6.2. Catalase Assay (CAT)

The catalase activity was assayed by the Catalase Assay Kit (Catalog No. 707002, Cayman Chemical, Ann Arbor, MI, USA). This method is based on the reaction of the enzyme with methanol in the presence of an optimal concentration of H_2_O_2_. The formaldehyde produced is measured colorimetrically with 4-amino-3-hydrazino-5-mercapto-1,2,4-triazole (Purpald) as the chromogen. Purpald forms a bicyclic heterocycle with aldehydes, changing from colorless to purple during oxidation. Changes in absorbance were recorded at 540 nm with a spectrophotometer. One unit of catalase is defined as the amount of enzyme that will cause the formation of 1.0 nmol of formaldehyde per minute at 25 °C. Experimental data is expressed as nmol formaldehyde/minute/milliliter.

#### 2.6.3. Glutathione Peroxidase Assay (GPx), Glutathione Reductase (GSH Rd) and Glucose-6-Phosphate Dehydrogenase (G6pdh) Assays

Glutathione peroxidase (GPx) activity was measured using the Glutathione Peroxidase Assay kit (Catalog No. 703102, Cayman Chemical, Ann Arbor, MI, USA). Glutathione peroxidase (GPx) oxidizes reduced glutathione (GSH) to glutathione disulfide (GSSG) while NADPH is oxidized to NADP^+^. There will be a decrease in absorbance at 340 nm during the oxidation of NADPH to NADP^+^. When GPx activity is rate-limited, GPx activity in the sample is proportional to the rate of decrease of A_340_. One unit of GPx is defined as the amount of enzyme that will cause the oxidation of 1.0 nmol of NADPH to NADP^+^ per minute at 25 °C.

Glutathione reductase (GSH Rd) activity was determined using the Glutathione Reductase Assay Kit (Catalog No. 703202, Cayman Chemical, Ann Arbor, MI, USA). The GSH Rd activity was measured by assaying the rate of NADPH oxidation. The oxidation of NADPH to NADP^+^ is accompanied by a decrease in absorbance at 340 nm. Since GSH Rd is at its rate limit concentrations, the rate of decrease of absorbance at 340 nm (A_340_) is proportional to the GSH Rd activity in the sample. One unit of GSH Rd is expressed as the amount of enzyme that will cause the oxidation of 1 nmol of NADPH to NADP^+^ per minute.

Glucose-6-phosphate dehydrogenase (G6PDH) activity was assayed by the Glucose-6-Phosphate Dehydrogenase Activity Assay Kit (Item No. 700300, Cayman Chemical, Ann Arbor, MI, USA). G6PDH is involved in the pentose phosphate pathway that produces NADPH. The product from the reaction of NADPH with fluorescent detector can be analyzed with an excitation wavelength of 530–540 nm and an emission wavelength of 585–595 nm. One unit is defined as the amount of enzyme that will catalyze the conversion of 1 nmol of G6P into 6PG and generates 1 nmol of NADPH per minute.

### 2.7. Oxidative Stress Measures

Carbonylated protein content was assayed by the Protein Carbonyl Colorimetric Assay Kit (Item No.10005020, Cayman Chemical, Ann Arbor, MI, USA) Protein carbonyl content was measured by the derivatization of protein carbonyl groups with 2,4-dinitrophenylhydrazine (DNPH) leading to the formation of hydrazine, which can be detected spectrophotometrically at 360–385 nm. Experimental data is expressed as nanomoles per milliliter of protein.

Thiobarbituric acid reactive species (TBARS) was measured by the TBARS Assay Kit (Item No. 10009055, Cayman Chemical, Ann Arbor, MI, USA). Lipid peroxidation is often used as an indicator of oxidative stress in cells and tissues. Malondialdehyde (MDA) is a naturally occurring end-product of lipid peroxidation that can be quantified through a controlled reaction with thiobarbituric acid to generate TBARS. The measurement of TBARS is a method for screening and monitoring lipid peroxidation by measuring absorbance of TBARS concentration at 530–540 nm. Experimental data is expressed as mol equivalent malondialdehyde µM/g protein.

### 2.8. Determination of 8-Hydroxy-2′-Deoxyguanosine in Mitochondrial DNA

Brain tissues were homogenized in 2 mL of SHE solution (0.25 M sucrose, 0.5 mM EDTA and 3 mM HEPES) and centrifuged at 800× *g* for 10 min at 4 °C. 1 mL of supernatant was centrifuged again at 9500× *g* for 10 min. Supernatant was discarded and 1 mL of SHE solution was added to the precipitate. The last two steps were repeated three times to give a mitochondrial suspension. The content of 8-hydroxy-2′-deoxyguanosine (8-OHdG) was determined using the Highly Sensitive 8-OHdG Check Enzyme-linked Immunosorbent Assay (ELISA) kit (Code. KOG-HS10E, Japan Institute for the Control of Aging, Shizuoka, Japan). Absorbance was measured at 450 nm and standard curve was used to calculate the 8-OHdG concentration expressed as ng per milliliter of DNA.

### 2.9. Determination of Dopamine Content

The concentration of dopamine was assayed using the Dopamine Research ELISA kit (Catalog NR. BA E-5300, Labor Diagnostika Nord, Nordhorn, Germany). The dopamine concentration was measured at 450 nm and expressed as nmol per liter of sample extracted.

### 2.10. RNA Extraction and Quality

Hippocampal tissues from the normal control group (*n* = 5 males, *n* = 5 females) and 1× ProBeptigen® group (150 mg/kg, *n*= 5 males, *n* = 5 females) were obtained and stored in RNA*later* stabilization solution (Thermo Fisher Scientific, Waltham, MA, USA). Tissues were homogenized using a TissueRuptor (Qiagen, Germantown, MD, USA) and total RNA extraction carried out using the RNeasy Microarray Tissue Kit (Qiagen, Germantown, MD, USA). RNA integrity was assessed using the Bioanalyzer 2100 (Agilent Technologies, Palo Alto, CA) and quantified using the NanoDrop^®^ ND-1000 Spectrophotometer (NanoDrop, Wilmington, DE, USA). A total of 20 arrays (five independent biological replicate × two gender groups × two treatment group) were generated.

### 2.11. DNA Microarray Chip-Based Whole Genome Brain Transcriptome Profiling

Global gene expression profiling was performed using the Agilent SurePrint G3 Gene Expression (8 × 60 K) v2 Microarray (Agilent Technologies, Santa Clara, CA, USA). There were 56,745 features in each array, inclusive of control probes. Each total RNA sample (100 ng) was amplified and labeled with the Low Input Quick Amp Labeling Kit (Agilent Technologies, Palo Alto, CA, USA) following the manufacturer’s instruction. The hybridization procedure was performed using the Agilent Gene Expression Hybridization Kit (Agilent Technologies, Design ID 074809, Santa Clara, CA, USA). Fluorescence signals of the hybridized Agilent microarrays were detected with the Agilent’s high resolution Microarray Scanner System (C-model, Agilent Technologies, Palo Alto, CA, USA). The Agilent Feature Extraction Software was used to obtain and process the microarray image files.

Data analysis was done using Genespring GX software. Comparison was made between SAMP8 untreated control vs. mice treated with ProBeptigen for each gender. Fold change and significance analysis was performed with a moderated *t*-test to identify the top significant genes. We also analyzed the differentially expressed genes (≥/≤ two-fold compared with control) with a *p* < 0.05 for enrichment/over-representation in functional categories i.e., gene ontologies and pathways using the 2 × 2 Fischer’s exact test implemented in the web-based Gene Ontology (GO) Consortium platform (http://www.geneontology.org/). Categories queried included the biological function, molecular function and cellular component gene ontologies and the reactome pathways.

### 2.12. Statistical Analysis

All data (except for microarray data) were analyzed using SPSS software version 19.0 (New York, United States) and are presented as means ± standard error of mean (SEM). The statistical significant differences were analyzed by a one-way analysis of variance (ANOVA) and the statistically significant group means were then compared using Duncan’s multiple range test.

## 3. Results

### 3.1. Effect of ProBeptigen on Changes in Body Weight, Food Intake and Locomotor Activity

There was no significant difference between control (No ProBeptigen) and ProBeptigen supplemented mice in terms of their initial and final body weights (Table 2). This shows that ProBeptigen supplementation did not affect both weight and daily food and water intake. No significant difference in the 10-minute-open field activity was observed between all groups in both male and female mice (Appendix A).

### 3.2. Effect of ProBeptigen on Total Grading Score of Senescence

SAMP8 mice show an irreversible advancement of senescence manifested by hair loss, increased skin coarseness, ulcers, lesions and lordokyphosis, as well as a loss of activity. This results in a high total grade score reflective of increased senescence (Table 3 and Table 4). Compared with control mice, supplementation of male mice with medium (150 mg/kg) or high dose (600 mg/kg) of ProBeptigen significantly lowered total grading scores. Low dose of ProBeptigen in male mice also reduced total grading scores, although not reaching statistical significance (Table 3). All three female mice groups fed with ProBeptigen at different doses (150 mg/kg, 300 mg/kg and 600 mg/kg) had lower grading score than the control group (Table 4). This demonstrates that ProBeptigen supplementation has a beneficial effect on delaying the aging process.

### 3.3. Effect of ProBeptigen on Cognition

#### 3.3.1. Effect of ProBeptigen on the Latency of Samp8 Mice in the Passive Avoidance Test

As shown in Figure 1, the latency of SAMP8 mice treated with ProBeptigen significantly increased compared to the control group for both male and female mice at 24 and 48 hours (*p* < 0.05). At 72 h after training, mice on ProBeptigen diet 300 mg/kg body weight (medium dose) and 600 mg/kg body weight (high dose) demonstrated significantly longer passive avoidance times than age-matched control group indicating that ProBeptigen supplementation could prolong long-term latency and lessen the aging-related cognitive deficits of mice, regardless of gender.

#### 3.3.2. Effect of ProBeptigen on the SAMP8 Mice in the Active Avoidance Test

The results from the avoidance trials showed that SAMP8 mice supplemented with ProBeptigen had improved learning and memory compared to the control group evidenced by the significant increase in the number of successful active avoidances (Figure 2). At the lowest dose (150 mg/kg body weight), the female mice showed significant learning improvement at Day 3, one day earlier than the male mice, which only showed improvement at Day 4. At higher doses i.e., 300 mg/kg body weight and 600 mg/kg body weight ProBeptigen, SAMP8 mice demonstrated a significantly greater number of successful avoidances compared to the control group for both male and female mice from Day 2 onwards.

### 3.4. Effect of ProBeptigen on Oxidation of Brain

#### 3.4.1. Effect of ProBeptigen on the Anti-Oxidative Enzymes in the Brain of Samp8 Mice

Administration of ProBeptigen (150, 300 and 600 mg/kg) significantly augmented the activities of antioxidative enzymes i.e., SOD, catalase and GPx in brain homogenates of six-month-old SAMP8 mice compared to age-matched controls (Figure 3A–C). These changes were concomitant with a significant increase in GSH reductase and G6PDH activities in all ProBeptigen fed groups (Figure 3D–E). This shows that long-term administration of ProBeptigen improved antioxidant capacity in the brains of SAMP8 mice.

#### 3.4.2. Effect of ProBeptigen on the Protein and Lipid Peroxidation in the Brain of Samp8 Mice

Compared with control mice, levels of oxidative stress markers were significantly decreased upon long-term supplementation of ProBeptigen at all doses. In the brain tissues of SAMP8 mice treated with ProBeptigen (low, medium and high doses), protein carbonyl, an index of protein oxidation [26], and lipid peroxide contents were reduced in both genders (Figure 4A,B). Consistent with these findings, TBARS levels were significantly decreased in all ProBeptigen-treated groups compared to the control (Figure 4C). TBARS is one of the indices of lipid peroxidation due to the high reactivity of thiobarbituric acid with the lipid peroxidation end product, malondialdehyde (MDA) [66]. The results suggest that the antioxidant defense in the brains of ProBeptigen-fed mice were better than that of the control group. Another measure of oxidative stress, 8-OHdG, was found to be remarkably reduced in the brain mitochondrial DNA of ProBeptigen supplemented groups regardless of dose or gender (Figure 4D).

#### 3.4.3. Effect of ProBeptigen on Brain Dopamine Concentration

As shown in Figure 5, the dopamine levels in the brains of ProBeptigen-fed mice were significantly higher than that of control groups. Dopamine is a monoamine/catecholamine neurotransmitter of the sympathetic nervous system, helping the body to cope with acute and chronic stress. Our results suggest that ProBeptigen administration may attenuate the decrease in the dopamine concentration in SAMP8 mice during aging to prevent the deterioration in learning and memory.

### 3.5. Transcript Profiling

Microarray analysis revealed 91 and 197 transcripts showing significant differences (≥/≤ 2-fold) in mRNA level in male and female mice on ProBeptigen diet (150 mg/kg) respectively, compared to control groups. For a list of all 288 genes and the fold changes, please refer to Appendix A. Many of the genes that are differentially expressed in the ProBeptigen-supplemented mice are functionally related. Pathway analysis using GO Terms (see Materials and Methods) revealed changes in several pathways as shown in Table 5.

We examined the genes that showed the most significant differential expression and noticed some major trends among these genes. ProBeptigen supplementation appears to modulate genes involved in glucose and lipid metabolism, social behaviors, olfactory learning, cognition in fear conditioning, brain development, inflammation and apoptosis. Some of the genes regulated by ProBeptigen supplementation in both male and female mice groups include *Mup-1, -3, -5, -9, -17, -21, -ps16, Ahsg* and *Alb* (shown in Figure 6).

Several genes that are regulated similarly in both genders by ProBeptigen supplementation compared to controls are major urinary protein 1 (*Mup1*; male: 16.45-fold, *p* = 0.02; female: 26.67-fold, *p* = 0.004), *Mup5* (male: 5.58-fold, *p* = 0.035; female: 4.3-fold, *p* = 0.0001), alpha-2-HS glycoprotein (*Ahsg*; male: 7.42-fold, *p* = 0.01; female: 15.47-fold, *p* = 0.004) and solute carrier family 47 member 1 (*Slc47a1*; male: 2.83-fold, *p* = 0.004; female: 2.87-fold, *p* < 0.001; Appendix A).

## 4. Discussion

In this study, long-term oral administration of ProBeptigen in three-month-old SAMP8 mice for 13 weeks did not affect the growth variables and food intake (Table 2). Motor activity and level of anxiety as observed in open field test (Appendix A) were also not affected. However, ProBeptigen at medium (300 mg/kg) and high (600 mg/kg) doses significantly lowered total grading score of senescence in SAMP8 mice (Table 3 and Table 4). The improvement in the signs of aging was not only limited to physical changes, but also behavioral performance as the administration of ProBeptigen at medium and high doses enhanced the latency of SAMP8 to enter the dark chamber in the passive avoidance test up till 72 hours compared to control groups (Figure 1). Lower dose of ProBeptigen (150 mg/kg) could retain the memory in SAMP8 mice up till 48 hours only. Consistently, SAMP8 mice on ProBeptigen regime demonstrated a better escape response over four days of training in the active shuttle avoidance compared to age-matched controls. These results suggest that long-term supplementation of ProBeptigen is able to delay the ageing process by effecting neurobehavioral and physical changes. In particular, ProBeptigen appears to enhance long-term memory and learning in the SAMP8 mice as evidenced by the improved passive avoidance.

Concomitant with the memory improvement, ProBeptigen modulates endogenous antioxidant defense systems in aged SAMP8 mice by increasing activities of antioxidant enzymes SOD, catalase and GPx compared to control mice. These changes were paralleled by augmented G6PD activity and increased GSH Rd levels (Figure 3). GSH Rd is an enzyme that catalyzes the reduction of glutathione disulfide (GSSG) to the sulfhydryl form glutathione (GSH), which in turn protects the brain from oxidative stress [67]. On the other hand, G6PD provides NADPH reductive power for ROS detoxification. It has been observed that increased G6PD activity in mice protects against ageing-associated functional decline and prolongs lifespan through augmented NADPH levels and lowered ROS-derived damage [68]. Compromised anti-oxidant defenses in SAMP8 mice could exacerbate the aging-related neurodegeneration as evidenced by decreased levels of GSH and antioxidant enzymes SOD, catalase and GPx in the brain [69,70,71]. ProBeptigen supplementation increased activities of GSH Rd and SOD in the brains of SAMP8 mice, which is in agreement with one of the studies showing a similar anti-oxidant and memory-enhancing effect of lotus seedpod proanthocyanidins (LPSC) [35]. Importantly, ProBeptigen supplementation did not affect the motor activity, anxiety level, response to shock or body weight. As it is known that anxiety, stress and motor activity could indirectly affect cognition [72,73,74], any effect by ProBeptigen on any of these parameters would have confounded the cognitive and behavioral results. 

The enhanced antioxidant activities may have resulted in a significant decrease in TBARS, lipid peroxide and protein carbonyl content (Figure 4A–C). In addition, ProBeptigen protects DNA from lipid peroxidation, as evidenced by a decline in 8-OHdG (Figure 4D). In SAMP8 mice, oxidative stress affects measures of protein oxidation, lipid peroxidation and oxidation-dependent changes in membrane protein conformation. Previous studies showed that the levels of lipid peroxides in two- to three-month old SAMP8 mice become significantly higher than in the SAMR1 mice, which are senescence resistant strains showing normal aging [32,75]. As the lipid composition of brain membranes plays an important role in memory in the SAMP8 mice, the higher amounts of ROS reactants such as MDA, TBARS and protein carbonyls detected in brain homogenates of SAMP8 compared to SAMR1 mice [27,70,76] might result in cognitive impairments. Taken together, our results imply that ProBeptigen is able to rescue the cognitive deficits via boosting the antioxidant defense to scavenge the free radicals and prevent further damage to essential cellular components (Figure 7).

The anti-oxidative properties of ProBeptigen may be due to the presence of potentially bioactive peptides and/or its amino acid composition. Two peptides (His-Val-Thr-Glu-Glu and Pro-Val-Pro-Ala-Glu) previously isolated from chicken essence were shown to exhibit antioxidant activities, including inhibition of linoleic acid autoxidation, DPPH radical scavenging activity, reducing power and the ability to chelate metal ions [77]. Thus, it is possible that the antioxidant properties of ProBeptigen are conferred by one or several bioactive peptides, or the complex interplay of the various amino acids and peptides. It would be interesting to determine if these two peptides and/or other bioactive peptides/amino acids may be responsible for the antioxidant activity of ProBeptigen. The mechanism and exact identification warrants further investigation.

Both aging and abnormal oxidative changes in the brain can give rise to neurochemical anomalies such as monoamine disturbances that lead to neuronal degeneration and cognitive deficits [78,79]. This is exemplified by the fact that the number of dopaminergic neurons and dopamine (DA) content in aged (8–10 months) SAMP8 mice are lower than age-matched SAMR1 mice [80]. Flood and colleagues [81] showed that dopamine agonists ameliorated impaired T-maze avoidance response in SAMP8. Administration of some antioxidant such as icariin, N-Test-butyl-alpha-phenylnitrone, vitamin C and E improved memory, protected the brain from oxidative stress and increased monoamine e.g., dopamine levels in aged SAMP8 mice [18,37,67]. Our results suggest that the improved cognitive performance by ProBeptigen administration may be due to enhanced levels of dopamine in the brains of SAMP8 mice (Figure 5).

A hydrolyzed chicken extract, ProBeptigen, contains an abundance of peptides and amino acids. Due to the high temperature and pressure during processing, ProBeptigen also forms a substantial amount (>7.5%) of diketopiperazines (DKPs) or cyclic dipeptides (Table 1). Diketopiperazines are natural compounds found endogenously in mammals [82,83] and in processed food such as chicken essence, roasted coffee and cocoa [84,85,86]. A few diketopiperazines and their derivatives have been found to be neuroprotective e.g., cyclo-prolylglycine (cyclo-(Pro-Gly)) expressed in rat brain and cyclo(L-Phe-L-Phe) in chicken essence whereby both have anti-amnesic activity effects [87,88,89]. Both DKPs are small and lipophilic molecules that can cross the blood–brain barrier (BBB) [89,90,91]. In particular, chronic oral administration of cyclo(L-Phe-L-Phe) reversed learning and memory impairment and increased extracellular levels of the cerebral monoamines serotonin, norepinephrine and dopamine in the medial prefrontal cortex of scopolamine-induced amnestic mouse model [89]. In the same study, the apparent permeability coefficient (Papp) of cyclo(L-Phe-L-Phe) was shown to be comparable to that of caffeine, which passes through BBB easily, using an in vitro BBB model. Other N-methyl derivatives of DKPs also have been reported to cross the blood–brain barrier easily [92]. It would be interesting to determine if any of the DKPs in ProBeptigen may be responsible for this observed increase in dopamine levels and concomitant improved cognitive behavior in the SAMP8 mice. Further studies on the region-specific expression of dopamine and other monoamines, as well as the identification of the bioactive in ProBeptigen warrants further investigation.

A previous microarray study looking at age-dependent gene expression changes of SAMP8 hippocampi showed that genes associated with the stress response and antioxidant systems are strongly affected during age-related cognitive impairment [63]. This is in agreement with our microarray data revealing that ProBeptigen supplementation regulates genes that are involved in similar pathways i.e., the oxidation–reduction process, energy metabolism (especially lipid), acute phase response and immune system (Table 5). Although many genes are differentially regulated in male and female mice, there are several genes that are regulated similarly in both genders by ProBeptigen supplementation. Major urinary protein 1 (*Mup1*) is one of the genes classified under the oxidation–reduction process and upregulated in both the ProBeptigen-supplemented male and female mice compared to their respective controls (Appendix A). Besides regulating energy expenditure, glucose and lipid metabolism [93,94], MUP-1 mediated increase in the expression of genes for mitochondrial biogenesis and more importantly, elevated mitochondrial oxidative capacity via increased activities of mitochondrial cytochrome c oxidase and citrate synthase [93]. This may explain the effect of ProBeptigen in reducing 8-OHdG, a marker of DNA oxidative damage in the brain. Another gene encoding major urinary protein, *Mup5* was also upregulated in the ProBeptigen-fed male and female mice (Appendix A). This upregulation may have contributed to the enhanced performance in active and passive avoidance tasks since *Mup5* has previously been shown to be downregulated in aged mice with cognitive impairments in contextual fear conditioning and spatial learning and memory [95]. 

Other genes that were upregulated include the alpha-2-HS glycoprotein (*Ahsg*) and solute carrier family 47 member 1 (*Slc47a1*; Appendix A). *Ahsg* encodes a multifunctional extracellular calcium regulatory glycoprotein involved in several processes including endocytosis and brain development. Interestingly, the *Ahsg* transcript has been found to be elevated in the discrete thalamic neurons of rats subjected to fear conditioned learning, suggesting a role for enhanced *Ahsg* expression in the learning and memory performance of the ProBeptigen supplemented mice [96]. *Slc47a1* encodes the solute carrier transporter on the neuronal and glial cell membrane to facilitate polyamine transportation. Natural polyamines such as spermine and spermidine play numerous roles including modulating ionic channels activity, protein synthesis and cell proliferation [97]. Particularly, intrahippocampal infusion of spermidine improved acquisition and memory consolidation in the inhibitory avoidance task [98,99,100]. Further validation of the ProBeptigen mediated gene expression changes and the role and mechanism of action of the encoded proteins remains to be elucidated.

## 5. Conclusions

ProBeptigen acted as an antioxidant to confer neuroprotection against the deleterious effects of ageing, particularly in learning and memory. ProBeptigen supplementation mediated hippocampal-specific gene expression changes, improved overall anti-oxidant defense and increased dopamine level, all contributing towards enhancing the cognitive performance in the accelerated aging SAMP8 mouse model.

## Figures and Tables

**Figure 1 nutrients-11-01870-f001:**
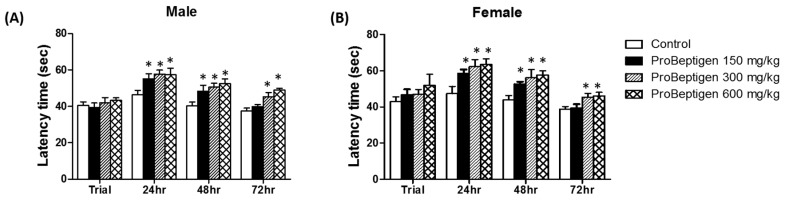
Passive avoidance ability of six-month-old SAMP8 mice fed saline (control) or ProBeptigen at different doses (150, 300 or 600 mg/kg/day) for 12 weeks from three months of age. Passive avoidance tests were performed on (**A**) male and (**B**) female groups of SAMP8 mice, *n* = 8 per group. Memory retention was tested 24, 48 and 72 h after the acquisition trial. Values were expressed as mean ± SEM and analyzed by one-way ANOVA. * indicates a significant difference in latency time (*p* < 0.05) compared to the control group.

**Figure 2 nutrients-11-01870-f002:**
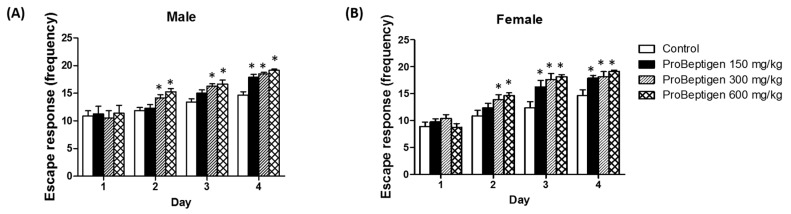
ProBeptigen enhanced learning and memory of a six-month-old SAMP8 mice. Active avoidance tests were performed on (**A**) male and (**B**) female SAMP8 mice that that were fed saline (control) or different doses of ProBeptigen (150, 300 or 600 mg/kg/day) for 13 weeks from three months of age (*n* = 8 per group). The tests were performed four times per day for four consecutive days. Values were expressed as mean ± SEM and analyzed by one-way ANOVA. * indicates significant difference in escape response time (*p* < 0.05) compared to the control group.

**Figure 3 nutrients-11-01870-f003:**
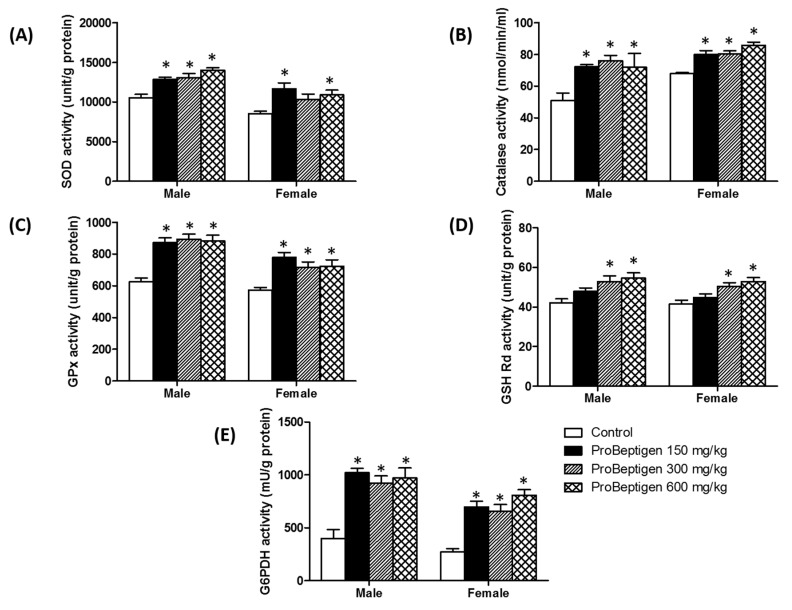
ProBeptigen enhanced antioxidant defenses in the brains of six-month-old SAMP8 mice. ProBeptigen activities of (**A**) superoxide dismutase (SOD), (**B**) catalase, (**C**) glutathione peroxidase (GPx), (**D**) glutathione reductase (GSH Rd) and (**E**) glucose-6-phosphate dehydrogenase G6PDH in both male and female SAMP8 mice fed different doses of ProBeptigen (150, 300 or 600 mg/kg/day) or not fed ProBeptigen (control) for 13 weeks (*n* = 8 per group). Values were expressed as mean ± SEM and analyzed by one-way ANOVA. * indicates a significant difference in the escape response time (*p* < 0.05) compared to the control group.

**Figure 4 nutrients-11-01870-f004:**
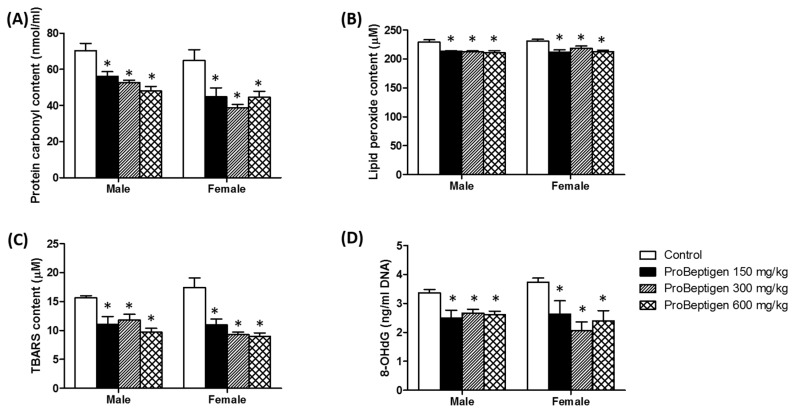
ProBeptigen decreased protein and lipid peroxidation in the brains of six-month old SAMP8 mice. Levels of (**A**) protein carbonyl, (**B**) lipid peroxide, (**C**) TBARS and (**D**) mitochondrial 8-OHdG in both male and female SAMP8 mice fed different doses of ProBeptigen (150, 300 or 600 mg/kg/day) or not fed ProBeptigen (control) for 13 weeks (*n* = 8 per group). Values were expressed as mean ± SEM. and analyzed by one-way ANOVA. * indicates a significant difference in the escape response time (*p* < 0.05) compared to the control group.

**Figure 5 nutrients-11-01870-f005:**
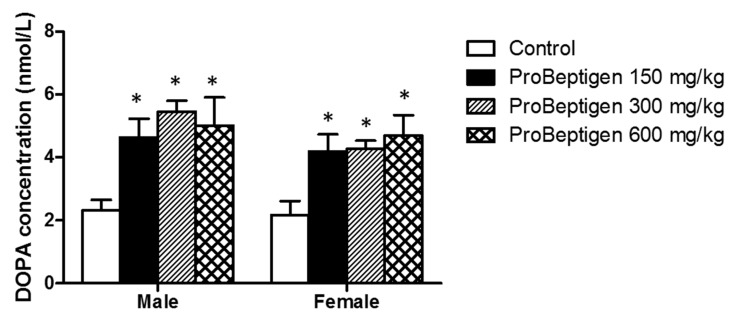
ProBeptigen regulated the concentration of dopamine in the brains of six-month-old SAMP8 mice. Both male and female SAMP8 mice were fed different doses of ProBeptigen (150, 300 or 600 mg/kg/day) or not fed ProBeptigen (control) for 13 weeks (*n* = 8 per group). Values were expressed as mean ± SEM and analyzed by one-way ANOVA. * indicates a significant difference in the escape response time (*p* < 0.05) compared to the control group.

**Figure 6 nutrients-11-01870-f006:**
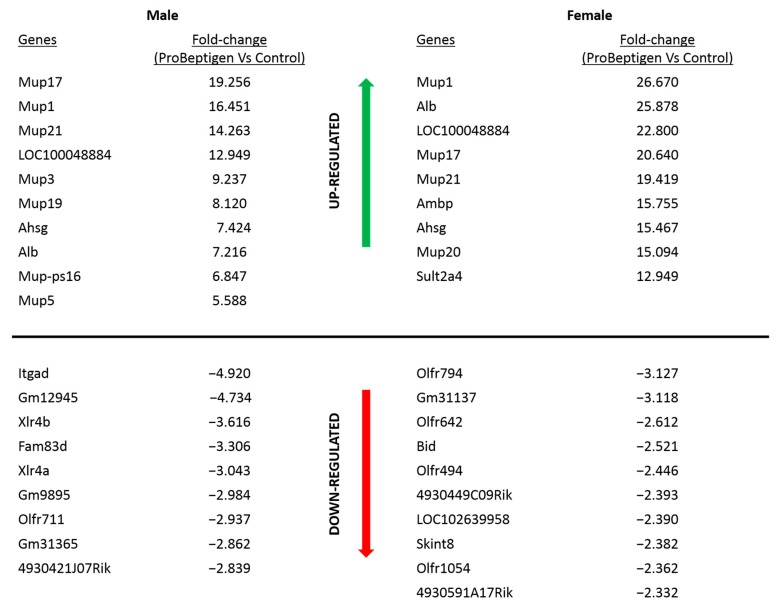
The top genes identified in the hippocampus of SAMP8 mice on ProBeptigen supplementation along with the fold changes (ratio of ProBeptigen against control group) indicated by green (up-regulation) and red (down-regulation) arrows.

**Figure 7 nutrients-11-01870-f007:**
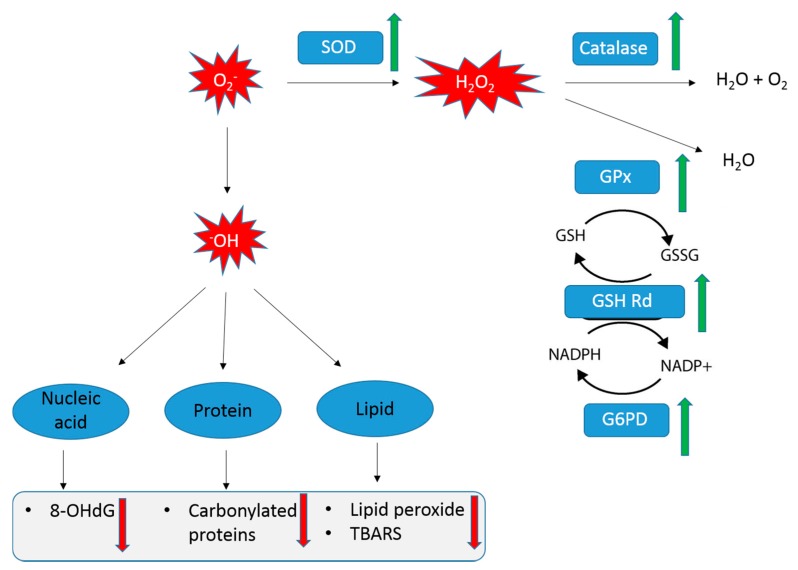
The oxidation pathways affected by ProBeptigen supplementation, suggesting a possible mechanism of action by which ProBeptigen enhances cognitive performance in SAMP8 mice. The concentration of the molecules modulated by ProBeptigen are indicated by green (increased) and red (decreased) arrows. ProBeptigen promotes the activities of antioxidant enzymes particularly SOD, catalase, GPx, G6PD and GSH Rd. This may lead to an increase in reducing agents GSH and NAPDH, augmenting the ability to scavenge free radicals such as reactive oxygen species (ROS). ProBeptigen may also protect the brain from protein and lipid peroxidation as well as reduce mitochondrial damage.

**Table 1 nutrients-11-01870-t001:** Nutritional content of chicken meat ingredient (ProBeptigen).

Ingredients	Amount (%)
Proteins and peptides	91.38
Free amino acids	4.23
Diketopiperazines	7.598
Carbohydrate	1.01
Lipid	1.64
Minerals	
Sodium	0.31
Potassium	0.34
Calcium	0.08
Magnesium	0.1
Chloride	0.53

**Table 2 nutrients-11-01870-t002:** Body weight and daily food and water consumption of six-month-old SAMP8 mice fed with or without ProBeptigen for 13 weeks.

Sex	Group	Body Weight (g)			Food Intake (g/day)	Water Consumption (ml/day)
		Initial	Final	Gain		
**Male**	**Control**	29.19 ± 0.49	29.71 ± 0.34	0.52 ± 0.43	4.61 ± 0.20	6.43 ± 0.18
	ProBeptigen (150 mg/kg)	29.91 ± 0.93	32.36 ± 1.39	2.45 ± 0.63	4.76 ± 0.16	6.57 ± 0.13
	ProBeptigen (300 mg/kg)	28.89 ± 0.89	30.63 ± 0.85	1.74 ± 0.79	4.98 ± 0.18	6.60 ± 0.22
	ProBeptigen (600mg/kg)	29.14 ± 0.67	30.24 ± 0.68	1.09 ± 0.50	4.51 ± 0.22	6.36 ± 0.33
**Female**	**Control**	25.86 ± 0.55	26.47 ± 0.57	0.61 ± 0.31	4.25 ± 0.16	5.32 ± 0.18
	ProBeptigen (150 mg/kg)	26.16 ± 0.96	27.00 ± 0.95	0.85 ± 0.74	4.21 ± 0.16	5.48 ± 0.16
	ProBeptigen (300 mg/kg)	26.25 ± 0.60	28.16 ± 1.11	1.91 ± 0.81	3.21 ± 0.10	5.40 ± 0.23
	ProBeptigen (600 mg/kg)	24.88 ± 0.83	26.44 ± 1.02	1.56 ± 0.57	4.28 ± 0.24	5.42 ± 0.12

Values were expressed as mean ± SEM and analyzed by one-way ANOVA. *n* = 8; data with * are significantly different (*p* < 0.05).

**Table 3 nutrients-11-01870-t003:** Total grading score of six-month-old SAMP8 male mice fed with or without ProBeptigen for 11 weeks

Group	Control	ProBeptigen (150 mg/kg)	ProBeptigen (300 mg/kg)	ProBeptigen (600 mg/kg)
Behavior				
Reactivity	0.50 ± 0.19	0.38 ± 0.18	0.13 ± 0.13	0.00 ± 0.00
Passivity	0.63 ± 0.18	0.38 ± 0.18	0.25 ± 0.16	0.13 ± 0.13
Skin and hair				
Glossiness	0.88 ± 0.23	0.75 ± 0.16	0.50 ± 0.19	0.63 ± 0.18
Coarseness	1.00 ± 0.19	0.88 ± 0.13	0.75 ± 0.16	0.75 ± 0.25
Hair loss	0.63 ± 0.26	0.50 ± 0.19	0.25 ± 0.16	0.13 ± 0.13
Ulcer	0.38 ± 0.18	0.25 ± 0.16	0.13 ± 0.13	0.13 ± 0.13
Eyes				
Periophthalmic lesion	0.38 ± 0.18	0.25 ± 0.16	0.13 ± 0.13	0.38 ± 0.18
Spine				
Lordokyphosis	0.88 ± 0.23	0.63 ± 0.26	0.50 ± 0.27	0.63 ± 0.18
Total	5.25 ± 0.59	4.00 ± 0.85	2.63 ± 0.68 *	2.75 ± 0.59 *

Values were expressed as mean ± SEM and analyzed by one-way ANOVA. *n* = 8; data with * are significantly different (*p* < 0.05).

**Table 4 nutrients-11-01870-t004:** Total grading score of six-month-old SAMP8 female mice fed with or without ProBeptigen for 11 weeks.

Group	Control	ProBeptigen (150 mg/kg)	ProBeptigen (300 mg/kg)	ProBeptigen (600 mg/kg)
Behavior				
Reactivity	0.38 ± 0.18	0.38 ± 0.18	0.25 ± 0.16	0.25 ± 0.16
Passivity	0.50 ± 0.19	0.25 ± 0.16	0.38 ± 0.18	0.25 ± 0.16
Skin and hair				
Glossiness	0.63 ± 0.18	0.38 ± 0.18	0.25 ± 0.16	0.38 ± 0.18
Coarseness	0.63 ± 0.18	0.50 ± 0.19	0.38 ± 0.18	0.50 ± 0.19
Hair loss	0.88 ± 0.30	0.38 ± 0.18	0.25 ± 0.16	0.25 ± 0.16
Ulcer	0.75 ± 0.25	0.50 ± 0.19	0.25 ± 0.16	0.25 ± 0.16
Eyes				
Periophthalmic lesion	1.00 ± 0.27	0.38 ± 0.18	0.63 ± 0.18	0.50 ± 0.19
Spine				
Lordokyphosis	0.50 ± 0.19	0.75 ± 0.16	0.50 ± 0.19	0.63 ± 0.18
Total	5.25 ± 0.56	3.50 ± 0.42 *	2.88 ± 0.55 *	3.00 ± 0.46 *

Values were expressed as mean ± SEM and analyzed by one-way ANOVA. *n* = 8; data with * are significantly different (*p* < 0.05).

**Table 5 nutrients-11-01870-t005:** Gene ontology (GO) terms and reactome pathways are listed with the corresponding *p*-value and the number of genes in the class represented on the array

GO Term/Pathway	Class Members on Array	*p*-Value
Heme binding	19	1.71 × 10^−14^
Regulation of insulin-like growth factor (IGF) transport and uptake by insulin-like growth factor	14	1.42 × 10^−8^
Carboxylic acid metabolic process	27	3.68 × 10^−8^
Oxidation-reduction process	28	7.37 × 10^−8^
Negative regulation of endopeptidase activity	12	4.68 × 10^−5^
Transition metal ion binding	24	8.14 × 10^−5^
Acute phase response	6	1.68 × 10^−3^
Innate immune system	22	4.13 × 10^−3^
Positive regulation of lipid catabolic process	5	1.39 × 10^−2^
Acylglycerol metabolic process	7	3.16 × 10^−2^
Regulation of lipoprotein lipase activity	4	4.83 × 10^−2^

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
