# Peer review of "Protective Effects of Hydrolyzed Chicken Extract (Probeptigen®/Cmi-168) on Memory Retention and Brain Oxidative Stress in Senescence-Accelerated Mice"

_nutrients, 2019, doi:10.3390/nu11081870_

Round 1

Reviewer 1 Report

This study aimed to clarify the effects of a chicken meat hydrolysate, ProBeptigen, on age-related cognitive function and oxidative stress in the brain by using senescence-accelerated prone (SAMP8) mice as a model.

The authors utilized total grading score of senescence, avoidance tests, antioxidant enzyme activity, dopamine levels, lipid peroxidation, a mitochondrial DNA content, and microarray analysis to demonstrate the results and suggested an anti-aging potential of ProBeptigen in improvement of learning and memory and antioxidant defense in the brain.

However, there are some minor revisions to concern before considering for publication in the Nutrients journal.

·         Materials and Methods: the data analysis is shown only for microarray analysis. There should also be the data analysis indicating software and statistical analysis for the remaining experiments/methods.

·         Table 2: there should be a footnote showing how the values were expressed and analyzed as demonstrated in the table 3.

·         Table 3: the footnote 2 is mentioned about Low dose, Medium dose and High dose of ProBeptigen; however, these words are not present in the table.

·         As indicated in 2.2 Animal treatment and tissue collection of Materials and Methods, the animals were treated with chow diet or ProBeptigen from 3 months old for 11-13 weeks (to 6 months old) then subjected to behavioral tests and dissected the brain tissues. However, in line 29 of the section 3.4.1 and the figure legends of Figure 3, 4 and 5, the results are elucidated “in the brains of 3-month-old SAMP8 mice”.

·         Figure 5: the size of the figure is too big as compared with the others.

·         Figure 6: the figure legend should be defined whether the fold change is the ratio between treated and control groups or log2 of the ratio (or else).

·         Line 99 – 122, these may be too long and suitable for the Introduction. The Discussion should mainly discuss and interpret the results/findings and their implications.

·         Line 154, please provide a reference or references of the sentence “As it is known that anxiety, stress, motor activity could indirectly affect cognition, ”

·         Line 163, (Wu et al., 2005) should be added in the References.

·         Line 178: there is no “Fig. 5A” in the manuscript.

·         Line 179 – 192: this paragraph is mentioned about the formation of diketopiperazines (> 7.5%) from ProBeptigen processing. If it is a result from this study, please insert it in the Results section, or if otherwise, please provide a reference or references.

·         Line 209, there is no Mup5 of female as referred to Figure 6.

·         Line 213, these is no data of fold change of Slc47a1 both male and female in the Results and supplementary file. New data/information of results should not be present for the first time in the Discussion.

·         Figure 7: there should be a paragraph to describe about this figure, or there should be more details in the figure legend of how the oxidation pathways affected by ProBentigen possibly enhance cognitive performance in mice, as illustrated in the figure.

·         Please check word spelling and consistent use of words.

Author Response

Thank you Reviewer 1 for your comments and valuable suggestions highlighting several areas needing correction and clarity. 

Please see the attachment for point-by-point responses and we hope that we are able to fully address your concerns. 

Reviewer 2 Report

In this study, the authors found that ProBeptigen supplementation enhanced the cognitive performance in the accelerated aging SAMP8 mouse model. Concomitant with the memory improvement, ProBeptigen increased antioxidant enzyme activity and dopamine level while reducing protein and lipid peroxidation and mitochondrial DNA damage in the brain. In addition, microarray analysis of hippocampus revealed several processes that may be involved in the improvement of cognitive ability by ProBeptigen, including heme binding, insulin growth factor regulation, carboxylic metabolic process, oxidation-reduction process and endopeptidase inhibition. Overall, the data are clear and of potential interest. Before publication, I would like the authors to address the following points.

1.      Are there any evidences which suggest that ProBeptigen improve cognitive performance via antioxidant activity? ProBeptigen may regulate neurogenesis and neuronal maturation other than the antioxidant defense system. The authors may need to check them.

2.      The authors discussed that the anti-oxidative properties of ProBeptigen may be due to the presence of potentially bioactive peptides and/or its amino acid composition. Is it possible that these peptides and/or amino acid pass through the blood brain barrier and into brain tissue followed by showing the anti-oxidatant activity? Is there any information regarding it?

3.      Which brain regions were used for the assessment of antioxidant enzymes?

4.      Please indicate methods for evaluation of carbonylated proteins.

5.      Long-term administration of ProBeptigen (for 11-13 weeks) improved the cognitive performance. Why do the authors choose 11-13 weeks? How about the effect of short-term administration on cognitive function?

6.      I cannot understand the following sentence in the 3-5th lines of the section 3.3.2. Please confirm whether it is right.

“This learning effect was more pronounced in female mice where improvement in mean number of successful avoidances at the lowest dose (150 mg/kg body weight) was seen by Day 3, compared to Day 4 in the male.”

7.      Please correct the following sentence in the 8th paragraph of the Discussion section.

“A few diketopiperazines and their derivatives have been found to be be neuroprotective e.g. cyclo-prolylglycine (cyclo-(Pro-Gly)) expressed in rat brain and cyclo(L-Phe-L-Phe) in chicken essence whereby both have anti-amnesic activity effects [80–82].”

Author Response

Thank you Reviewer 2 for taking the time to review the manuscript and for your comments and valuable suggestions. 

Please see the attachment for the point-by-point responses and we hope this would fully address the concerns. 
